# Identification of Novel Biomarkers of Spinal Muscular Atrophy and Therapeutic Response by Proteomic and Metabolomic Profiling of Human Biological Fluid Samples

**DOI:** 10.3390/biomedicines11051254

**Published:** 2023-04-23

**Authors:** Megi Meneri, Elena Abati, Delia Gagliardi, Irene Faravelli, Valeria Parente, Antonia Ratti, Federico Verde, Nicola Ticozzi, Giacomo P. Comi, Linda Ottoboni, Stefania Corti

**Affiliations:** 1Dino Ferrari Centre, Department of Pathophysiology and Transplantation (DEPT), University of Milan, 20122 Milan, Italy; 2Stroke Unit, Fondazione IRCCS Ca’ Granda Ospedale Maggiore Policlinico, 20122 Milan, Italy; 3Neurology Unit, Fondazione IRCCS Ca’ Granda Ospedale Maggiore Policlinico, 20122 Milan, Italy; 4Laboratory of Neuroscience, Department of Neurology, IRCCS Istituto Auxologico Italiano, 20095 Milan, Italy; 5Department Medical Biotechnology and Translational Medicine, University of Milan, 20100 Milan, Italy; 6Neuromuscular and Rare Diseases Unit, Department of Neuroscience, Fondazione IRCCS Ca’ Granda Ospedale Maggiore Policlinico, 20122 Milan, Italy

**Keywords:** antisense oligonucleotide, cerebrospinal fluid, proteome, metabolome, nusinersen, spinal muscular atrophy

## Abstract

Spinal muscular atrophy (SMA) is a neuromuscular disease resulting from mutations or deletions in *SMN1* that lead to progressive death of alpha motor neurons, ultimately leading to severe muscle weakness and atrophy, as well as premature death in the absence of treatment. Recent approval of SMN-increasing medications as SMA therapy has altered the natural course of the disease. Thus, accurate biomarkers are needed to predict SMA severity, prognosis, drug response, and overall treatment efficacy. This article reviews novel non-targeted omics strategies that could become useful clinical tools for patients with SMA. Proteomics and metabolomics can provide insights into molecular events underlying disease progression and treatment response. High-throughput omics data have shown that untreated SMA patients have different profiles than controls. In addition, patients who clinically improved after treatment have a different profile than those who did not. These results provide a glimpse on potential markers that could assist in identifying therapy responders, in tracing the course of the disease, and in predicting its outcome. These studies have been restricted by the limited number of patients, but the approaches are feasible and can unravel severity-specific neuro-proteomic and metabolic SMA signatures.

## 1. Introduction

Spinal muscular atrophy (SMA) is a neurodegenerative disorder caused by mutations in *SMN1* that determine a reduction in SMN protein [1,2] and a resulting loss of alpha motor neurons (MNs) in the brainstem and spinal cord, followed by progressive muscle weakness and atrophy, as well as early death [1,2]. Three types of pediatric SMA are recognized (types 1 to 3), as well as two less frequent types, prenatal (type 0) and adult (type 4) [1,2].

The full-length *SMN* mRNA is translated into a ubiquitously expressed 38-kDa protein [1,2]. The SMN protein is located inside the cytoplasm of various types of cells and in nuclear punctuated structures within the nucleus called gems. In fibroblasts from SMA patients, more gems are detected, less severe is the disease [1,2]. In neurons, SMN is present in axonal granules and moves bidirectionally at a rapid rate [1,2]. Recent studies have provided further insight into the role of SMN in cellular compartments and its association with disease progression in SMA patients [1,2].

Although SMN is ubiquitously expressed, motor neurons (MNs) are primarily affected by decreased expression [3,4]. Nonetheless, SMA seems to affect features in multiple tissues and this must be considered in a therapeutic perspective [5].

SMN is involved in the synthesis of uridine-rich spliceosomes and RNA, essential activities for the proper function of all eukaryotic cells [3,4] (Figure 1). Furthermore, in different type of cells, SMN is key in maintaining spliceosomal small nuclear ribonucleoprotein (snRNP) assembly for major (U2-dependent) and minor spliceosomes [6]. When SMN is absent, the U12 spliceosome is defective and causes an abnormal splicing of a relatively small number of genes. One of them is *TMEM41B*, which encodes for stasimon, a protein found in the endoplasmic reticulum (ER) whose function is largely unknown. The splicing profile of stasimon is disrupted in a mouse model of severe SMA. However, the exogenous administration of full-length stasimon by gene transfer prevents only the loss of proprioceptive inputs, but not of MNs. SMN contributes also to the assembly of mRNA-binding protein complexes, such as messenger ribonucleoprotein transport granules (mRNPs) at the distal ends of neurons, facilitating mRNA transport. However, in SMA patients, reduction in mRNP granules is still debated. In addition, SMN plays a role in gene transcription and protein translation [2].

Moreover, SMN deficiency affects crucial cellular processes such as cytoskeletal actin dynamics, synaptic vesicle release, and endocytosis [7]. These pathways play a significant role in the pathogenesis of SMA, as indicated by the discovery of SMA disease-modifying genes, such as *PLS3* and *NCALD*, which operate independently of SMN expression. Plastin-3, an actin-binding protein encoded by *PLS3*, has been shown to reduce disease severity in female SMA patients, whereas neurocalcin-δ, a calcium-binding protein encoded by *NCALD*, has a protective effect when its expression is reduced. The expression of plastin-3 in SMA mice with *NCALD* suppression has been shown to enhance endocytic neuromuscular junction pathways and improve aspects of the disease. Nevertheless, therapeutic targeting of *PLS3* and *NCALD* expression can be challenging due to their pleiotropic effects [7].

Further, SMN deficiency in MNs and muscles may result in mitochondrial reactive oxygen species generation [8,9]. According to a muscle spectroscopy study, SMA type 3 or 4 may have impaired mitochondrial ATP synthesis, resulting in an increase in blood lactate during exercise and impaired accumulation of intramuscular inorganic phosphate. Overall, several functional defects that vary among cell types and/or developmental stages may contribute to the disease course of SMA [2].

Patients with SMA have at least one copy of the paralogous gene *SMN2*. Although *SMN2* is transcribed at the same level as *SMN1*, it cannot completely compensate for *SMN1* loss because most of its mRNA lacks exon 7, which is essential for synthesizing functional full-length SMN protein [10,11,12]. An inverse relationship has been found between symptoms severity and *SMN2* copy number [2].

There are three types of pediatric SMA (types 1 to 3), and two less frequent types, prenatal (type 0) and adult (type 4). Despite this classification considers the age at diagnosis and the motor milestones achieved in untreated patients, it does not account for the phenotypic changes resulting from the advent of new therapies [2,11,12].

FDA and the European Commission have recently approved three innovative treatments for SMA that aim at increasing SMN protein levels and have demonstrated clinical efficacy and safety for a period of at least 12 months: nusinersen, an antisense oligonucleotide (ASO) that is administered intrathecally in the cerebrospinal fluid (CSF) and targets *SMN2* mRNA [13,14,15]; onasemnogene abeparvovec, a gene therapy based on intravenously injected adeno-associated virus (AAV) vector for *SMN* [2,16]; and risdiplam, a small molecule that is taken orally [17,18] and increases levels of full-length SMN by facilitating the inclusion of exon 7 in *SMN2* pre-mRNA transcripts.

Nusinersen is an 18 nucleotide-long 2′-O-(2-methoxyethyl) phosphorothioate ASO that targets intron 7 on *SMN2* pre-mRNA splicing silencer to promote exon 7 inclusion. The *SMN2* pre-mRNA splicing modification can compensate for the molecular consequences of genetic defects in *SMN1* by restoring full-length SMN protein levels [19,20]. Due to its incapability of crossing the blood-brain barrier, nusinersen must be administered intrathecally [20]. The treatment is effective for children with infantile-onset (SMA type 1) and later-onset (SMA type 2) SMA [13,14,15], as well as for adults [21]. However, improvements are limited when the symptoms linger for an extended period of time and few data are available on long-term therapeutic effects in adults.

No biomarkers have yet been identified either in the serum nor CSF of SMA patients that can be used to reliably assess disease progression or predict individual responses to therapy. With the availability of more therapeutic options, this will become more relevant [22,23].

Recently, researchers proposed and investigated several biomarkers for patients affected by SMA including molecular biomarkers, such as soluble *SMN* mRNA and protein expression; neurofilament light (NFL) and heavy (NFH) chain proteins; creatine kinase and creatinine levels; structural biomarkers, such as imaging studies and neurological examination results; and clinical indicators [22,23].

Of particular interest is the monitoring of NFL/NFH levels in the serum and CSF of SMA patients, as they are thought to correlate with disease progression. Studies involving NFL/NFH measurements in SMA patients suggest an association between elevated baseline values at diagnosis and poorer prognosis post-treatment [24]. Therefore, measuring these two markers may provide insights into the efficacy of the treatment intervention and help to inform on decisions regarding patient prognosis or management.

As regards other potential traceable biomarkers, it is important to mention that the dysregulation in microRNA (miRNA) biogenesis and metabolism can result from reduced levels of SMN protein in SMA patients. To identify potential biomarkers in SMA, Bonanno et al. [25] investigated the levels of four myomiRs (miR-133a, -133b, -206, and -1) in serum samples from 21 infantile SMA patients at baseline and after 6 months of nusinersen therapy [25]. The results of this study indicated that myomiR concentrations decreased after nusinersen therapy and interestingly lower levels of miR-133a were linked to better clinical outcomes. These results indicate that myomiRs may be helpful biomarkers for tracking the development of the disease and the effectiveness of treatment in SMA patients. Although these results are promising, additional studies are needed to confirm the role of myomiRs as potential biomarkers in SMA.

Another field with prognostic potential in SMA is the immune system as SMA patients exhibit also immune system abnormalities, although the underlying nature of these deficiencies remains largely unexplored. To gain insight into these aspects and determine if they can serve as potential biomarkers, Bonanno et al. [26] conducted a study to evaluate innate and adaptive immune response patterns in type 1–3 SMA patients before and after nusinersen treatment compared to healthy adult controls and pediatric reference ranges. At baseline, SMA patients had significantly higher levels of various inflammatory cytokines, including IL-1β, IL-4, IL-6, IL-10, IFN-γ, IL-17A, IL-22, IL-23, IL-31, and IL-33, in both serum and CSF samples [26]. However, after 6 months of nusinersen administration, no significant differences in serum or CSF were detected, although a reduction for some of these cytokine levels was observed in both adult and pediatric SMA patients. Furthermore, higher concentrations of specific cytokines at baseline were associated with poorer functional outcomes after treatment in pediatric patients. Conversely, an increase in other cytokines correlated with improved outcomes. These results suggest that monitoring pro- and anti-inflammatory mediators may be a useful approach for tracking disease progression and therapeutic response in SMA, although these markers have not been conclusively proven to be suitable and reliable [22,23].

In addition, no biomarker has been identified to date that can provide accurate indications of potential early therapeutic response in adolescent and adult SMA patients.

Soluble biomarkers are handy and promising tools for evaluating a treatment response. Indeed, serum or plasma can be obtained relatively easily and in a non-invasive manner through peripheral vein puncture. CSF is also a useful biofluid to screen for SMA bio-analytes because it is directly in contact with affected tissues (lower brain stem and spinal cord) and, while nusinersen is administered into the patient’s CSF, the latter can be collected without further invasive procedures. Moreover, the identification of nusinersen-dependent modifications in the molecular composition of the CSF may contribute to the knowledge on the underlying disease mechanisms. Furthermore, CSF analysis can reveal information about drug safety and potential side effects.

Technological advances in proteomic/metabolomic research have enabled omics analyses of a variety of biological fluids, providing novel insights into disease mechanisms and potential biomarkers [27] to inform on the therapeutic response, clinical trial design, and new therapeutic targets and approaches.

We present a critical review of studies that have performed untargeted proteomic and metabolomic analyses of blood and CSF from SMA patients (Figure 2 and Table 1), treated and untreated.

A literature search was performed in PubMed using keywords “Spinal Muscular Atrophy” and “omic or proteomic or metabolomic” to identify articles published in English between 1 January 2000 and 31 March 2023. Inclusion criteria were all studies on samples from human biological fluids from SMA patients. Exclusion criteria were non-human samples or samples that were not biological fluids (Figure 3).

## 2. Blood Proteomic Studies of SMA

### 2.1. Identification of Proteins, Metabolites, and Transcripts in Body Fluids of Untreated Patients: Biomarkers for SMA Study (BforSMA)

Finkel et al. carried out the first unbiased omics approach to perform quantitative discovery of candidate biomarkers in plasma from SMA patients. In this cross-sectional study (Biomarkers for SMA, BforSMA), 108 children with SMA (type 1–3, age 2–12 years) and 22 age-matched controls were studied to detect bio-analytes associated with functional capability and SMA type [28]. The new set of markers was evaluated in relation to the modified Hammersmith Functional Motor Scale (MHFMS), which measures disease severity and disability (Figure 4).

Proteomics, multidimensional liquid chromatography, and eight-plex iTRAQ labeling with mass spectrometry (MS) were used to analyze the peptides.

Concurrently, a multi-platform metabolomics study was carried out on plasma organic lipid acids using a QStar Elite quadrupole time-of-flight instrument (MDS/SCIEX). Multiple reaction monitoring (MRM) allowed the identification of 42 proteins.

A total of 200 analytes were associated with MHFMS scores, with plasma proteins showing the highest correlation. Three analytes, dipeptidyl peptidase IV (DPP4), osteopontin (OPN), and tetranectin (TET), which are related to muscle remodeling, positively correlated with MHFMS scores, whereas two others, fetuin-A and vitronectin (VTN), had negative correlations. However, the reason why when the former increases during muscle wasting, such as with bedrest [34], while the latter decreases as MHFMS improves, is unknown.

Rather than being directly caused by SMN levels or by the disease itself, these molecular alterations may represent compensatory changes that indirectly contribute to the overall features of the disease.

### 2.2. DPP4

DPP4 (CD26) is a transmembrane enzyme found on the surface of most cell types. It cleaves X-proline dipeptides from polypeptides, such as growth factors, chemokines, and cytokines producing novel bioactive molecules that can contribute to the activation of inflammatory cells. DPP4 acts also on incretins such as GLP-1, which are essential for controlling blood sugar levels. Currently, CD26 inhibitors are being investigated as potential therapeutics for diabetes mellitus, immune-mediated diseases, and cancer [35].

The activity of DPP4 is reduced during muscle wasting [36] but in murine skeletal muscles DPP4 inhibition results in anabolic effects [37] and, similarly in humans, muscle mass and the muscle/fat ratio are increased [38]. The molecular mechanism by which the variation in DPP4 activity affects muscles is unclear [39].

Remarkably, recent studies have shown that DPP4 modulators exhibit significant neuroprotective properties and can reverse the pathological mechanisms of Alzheimer’s disease (AD) in AD in vivo models [40].

### 2.3. OPN

OPN is a multifunctional glycoprotein that is synthesized by multiple cell types and is present in both intracellular (iOPN) and secreted (sOPN) forms. The various forms of OPN are implicated in specific biological processes [41].

OPN interacting with integrins regulates cell adhesion, migration, immune response, and tissue regeneration, playing a key role in inflammatory, degenerative, autoimmune, and oncological diseases. As a cytokine, OPN enhances interferon-gamma production and reduces IL-10, modulating a type I immunity. Interestingly, OPN modulates inflammation associated with muscular damage and is required in the early phases of muscle regeneration, as in dystrophinopathy [42]. However, muscle regeneration may be impaired when OPN is expressed chronically, as in Duchenne muscular dystrophy subjects [42].

Interestingly, in several neurological conditions, recent studies have suggested that OPN may play a role in tissue repair and regeneration likely thanks to its distinct functional domains able to facilitate interactions among cells and between cells and the extracellular matrix. Despite these contradictory features of OPN, understanding its mechanisms could be crucial to the development of targeted therapies for neurological disorders [43].

### 2.4. TET

The calcium-binding protein TET belongs to the C-type lectin family. It is primarily present in the extracellular matrix and serum during tissue regeneration, cancer, and development by activating proteases and growth factors and regulating extracellular matrix proteolysis to protect muscles, bones, and the circulatory system. Notably, TET has been linked to the regeneration and myogenesis of skeletal muscles [44].

TET concentrations in serum are reduced in cancer, inflammatory diseases, and coronary artery disease. Knock out mice for TET present impaired wound healing, bone fracture repair, and Parkinson’s disease (PD)-like symptoms as they age [45]. Of note, TET is considered a promising biomarker for PD given its reduced expression in the CSF of PD patients and because it has a neuroprotective role on dopaminergic neurons in culture. Thus, TET has the potential to be used as biomarker or therapeutic target in patients with neurodegenerative disorders [46].

### 2.5. Fetuin-A

Fetuin-A is a multifunctional protein originally identified in the bovine fetal serum and shown to be involved in the transport of fatty acids in the blood. It consists of two chains linked by a disulphide bridge. It is primarily expressed by mature hepatocytes and embryonic cells, with minor amounts also detected in adipocytes and monocytes. Fetuin-A has also neuroprotective and anti-inflammatory features [47]. Its levels are diminished upon injury, infection, and inflammation, raising the risk of metabolic disorders, cardiovascular disease, and insulin intolerance. Overall, the dysregulation of fetuin-A, which is essential for preserving the body’s homeostasis, is associated with a number of diseases [47].

### 2.6. VTN

VTN, a glycoprotein found in the plasma and extracellular matrix, regulates cell attachment, spreading, and migration and is important to maintain pluripotent stem cells and promote their expansion [48]. In addition to protecting the brain, VTN also regulates axon size, promotes neurite extension, reduces blood-brain barrier permeability, and regulates neural differentiation and axonal growth. Currently, how VTN functions in neurodegenerative diseases is not clear [48].

### 2.7. Multi-Analyte Panel Assay (SMA-MAP) of BforSMA

Twenty-seven BforSMA protein analytes were selected on the basis of associations with motor abilities and non-motor outcomes for validation in a multi-analyte panel assay (SMA-MAP) [49]. Further investigations of these analytes were conducted in plasma from SMN-Delta7 mice, showing that a few markers correlate with murine SMA symptoms and postnatal SMN rescue. The levels of DPP4, fetuin-A, OPN, TET, and VTN in SMA animals differed significantly from wild-type controls but returned to normal level after morpholino ASO therapeutic treatment [50]. All analytes but fetuin-A paralleled the SMN expression in the brain, spinal cord, liver, and quadriceps muscles. In addition, VTN played an important role because it correlated with the amplitude of compound muscle action potential (CMAP), a parameter of motor function. This suggests that these analytes can be used to assess treatment responses. Further, plasma levels of TET and VTN showed significant changes in 90-day-old mice as well, despite the decreased concentrations of ASO at this later time point. Thus, both proteins may serve as pharmacodynamic biomarkers for monitoring long-term therapeutic responses and for predicting need for re-treatment. A further validation of the SMA-MAP panel was carried out in the NeuroNEXT SMA child biomarker study by comparing serum samples from healthy and SMA infants of similar age [51]. The levels of five analytes (DPP4, TET, cadherin-13, cartilage oligomeric matrix protein, and insulin-like growth factor binding protein 6) were lower in SMA patients, whereas two analytes (myoglobin and chitinase-3-like protein 1) were elevated. SMA and healthy infants were distinguishable by these markers both at baseline and at later time points. In both cohorts, fibrin-1C, tenascin-X, thrombospondin-4, cartilage oligomeric matrix protein, complement component C1q receptor, and tyrosine kinase had negative correlations with age. Furthermore, blood concentrations of many proteins diminished with age, with the exception of apolipoprotein B in healthy infants and amyloid P in SMA patients.

It is difficult to draw a general conclusion on the SMA-MAP panel, although seven of them may have the capacity to differentiate subjects with SMA from healthy controls, supporting their role as a prognostic panel.

### 2.8. Cerebrospinal Fluid Proteomic Profiles in SMA Patients Treated with Nusinersen

#### CSF Proteomic by Mass Spectrometry

The first CSF proteomic study by MS in SMA patients (type 2 or 3; *n* = 10) aimed to correlate CSF analytes with clinical outcomes (Hammersmith Functional Rating Scale Expanded (HFMSE)) prior to and after nusinersen therapy [29]. All patients in the cohort had SMA type 3a or 3b, two subtypes of SMA type 3, except one subject with SMA type 2. Age- and gender-matched controls (*n* = 10) were also analyzed. Controls had only one lumbar puncture; conversely, SMA patients were treated with nusinersen and underwent several lumbar punctures. CSF samples taken before therapy (T0) and after 10 months of therapy (T10) were used for analysis. The evaluation period was selected to ensure constant steady-state drug levels after a loading dose and one maintenance dose according to nusinersen posology (Figure 5).

Peptide analysis was performed using a Thermo Scientific QExactive HF X mass spectrometer (Waltham, MA, USA) using data dependent measurement strategies.

Using mass spectrometry analysis of the CSF proteins, principal component analysis (PCA) identified two distinct clusters for SMA individuals, as well as a cluster for controls.

Forty-two proteins were differentially detected between SMA patients and age-matched controls at baseline.

Protein deglycase DJ-1, also known as PD protein 7 (PARK7) and the most up-regulated protein in SMA CSF at baseline (T0), is important in mediating neuroprotection in neurodegenerative disorders [52] and several studies have linked DJ-1 to PD. Studies have shown that DJ-1 functions as a chaperone under conditions of oxidative stress, which prevents the aggregation of alpha-synuclein [53]. DJ-1 is a chaperone sensitive to oxidative stress and able to protect neurons from the latter and from cell death. It has been shown that changes in the redox status of the cell determine DJ-1′s conformational changes, allowing DJ-1 interactions with other proteins and compounds. Indeed, DJ-1 inhibits the aggregation of alpha-synuclein and responds to changes in the redox state of the cell to preserve cellular homeostasis and avoid neuronal death [53].

A well-established protective role of PARK7/DJ-1 has been demonstrated in ischemic stroke and neurodegenerative diseases such as PD and AD. A reduction in DJ1 has been reported in the CSF of patients with PD [54]. In addition, PARK7/DJ-1 plays a critical role in maintaining the gut microbiome and in regulating intestinal inflammation. The connection between gastrointestinal inflammation and neurodegenerative disorders such as PD supports its role in neurodegeneration.

Within SMA patients, regardless of treatment, two major clusters (APC1 and APC2) have been observed and associated with age of subjects, not with motor functions. When the two clusters were compared, 268 proteins were found to be up-regulated in APC1 and 109 proteins upregulated in APC2 [29].

Two different molecular signaling networks were enriched in each CSF proteome cluster. The CSF proteome cluster linked with younger patients (APC1; 28.2 ± 4.6 years) was related to molecular pathways associated with neuroregeneration. The cluster associated with older patients (APC2) had up-regulated signaling networks associated with neurodegeneration. A significant and novel finding of this study was that differential expression in multiple pathways pinpointed clinical characteristics of the patients.

Compared with baseline, nusinersen treatment decreased the discrepancy between the SMA and control CSF proteomes, but cluster assignments remained unchanged. Overall, nusinersen treatment did not robustly affect the expression of any CSF protein. The authors speculated that variation in the neurochemical composition of CSF may be difficult to be detect during the short-evaluated period because of the slow progression of SMA in this cohort. Whether these results are due to the robustness of intra-individual CSF proteome fingerprints or only to the small sample size is unclear. Nonetheless, this was the first study to analyze the effects of nusinersen on protein expression in adults with SMA [29].

As a result of nusinersen administration, Kessler et al. [29] found intra-individual differences in protein modulation, resulting in the generation of two protein clusters (DPC1 and DPC2); patients in the DPC1 cluster had improved motor functions with nusinersen treatment, whereas patients in the DPC2 cluster remained stable. Patients with stable HFMSE scores in cluster DPC2 had higher levels of neuronal pentraxin-1 (NPTX1), semaphorin 7A (SEMA7A), carboxypeptidase E (CPE), and Collagen VI A (COL6A1); whereas, cadherin 18 (CDH18) levels were lower in five out of seven patients in cluster DPC1. Notably, there is a link between these proteins and neuronal or muscular functions.

### 2.9. NPTX1

NPTX1 plays a role in synaptic regulation [55]. CSF NPTX1 and NPTX2 levels are decreased in genetic forms of frontotemporal dementia and they may represent a broad class of biomarkers for neurodegenerative diseases [56].

### 2.10. SEMA7A

SEMA7A is involved in axonal guidance and belongs to a family of membrane-bound and soluble proteins that participate in axonal guidance as well as immunomodulation [57]. CSF levels of SEMA7A are significantly reduced in multiple sclerosis converters, which may be linked to its role as inhibitor of T-cell activation [58].

Semaphorins are secreted proteins that bind to neuropilins and plexins to exert either repelling or attractive functions. According to a recent study, semaphorin 3B (SEMA3B) is secreted into the CSF by the choroid plexuses and influences the proliferation and division orientation of neuronal progenitors particularly during development [59]. The contribution of soluble semaphorins to cortical progenitor cell activity remains an unanswered question [60]. Semaphorins are interesting emerging clinical biomarkers and therapeutic targets in different human pathological conditions including cancer [61]. The effects of semaphorin variations in the CSF in SMA and other neurological disorders should be better investigated.

### 2.11. CPE

CPE, also called neurotrophic factor-1 (CPE-NF-1), plays a central role in the synthesis of a wide range of peptide hormones and neurotransmitters, including stress and development factors in the central nervous system (CNS) [62]. CPE-NFα1 is reduced in patients with cognitive decline [62,63].

### 2.12. COL6A1

COL6A1 is primarily associated with the extracellular matrix including CNS and skeletal muscle; its mutation has been identified as the cause of different myopathies [63].

### 2.13. CDH18

CDH18 has an important role in the organization of the neural circuitry and in synapse maturation [64]. CDH18 is elevated in CSF of pregnant women [65]; even if the biological meaning of its variations is unknown, this finding suggests that CDH18 is a detectable metabolite that varies with the patient’s condition.

The reason for the higher levels of neuroprotective proteins detected in patients who did not respond to nusinersen and remained clinically stable is unclear. Further, despite candidate proteins having mainly neuroregenerative functions, the extent to which they can be used as potential biomarkers for SMA patients remains to be assessed. Longitudinal studies of CSF proteomes that monitor protein modifications are required.

Overall, despite the limitations associated with the small number of evaluated samples (*n* = 10), Kessler et al.’s study [29] found some differentially expressed candidate proteins that may contribute to the pathogenesis of SMA, to the beneficial therapeutic effects of nusinersen, and that may be linked to potential side effects of long-term intrathecal administration. New SMA biomarkers will probably be found and confirmed by expanding the study’s participant pool and monitoring the participants for extended periods of time. Since SMA is a progressive condition with a quite long survival time in populations with adult onset, it seems reasonable that international, multicenter, long-term studies are planned and carried out to undertake CSF proteomic studies of SMA patients using standardized protocols. Providing the first proof-of-concept proteomic data on patients with this slowly progressive neurodegenerative disease, Kessler et al. demonstrate the feasibility of CSF proteomic analysis. Molecular signaling networks that correlate with clinical outcome parameters can be used to identify relevant, potentially predictive biomarkers [29].

### 2.14. CSF Proteomic Profiles with 2D-PAGE

To determine whether the CSF proteomic responses upon nusinersen treatment could be clearly identified in SMA type 1 infants (*n* = 10) and to monitor disease progression, Bianchi et al. used a different proteomic approach [30]. To identify significantly different protein patterns, the authors applied SDS-polyacrylamide gel electrophoresis (2D PAGE) and MALDI-TOF MS (Autoflex Speed MS, Bruker Daltonics, Billerica, MA 01821, USA); Western blots were used for validation. All patients received nusinersen following the recommended protocol on days 0, 14, 28, and 63 (loading phase), and CSF samples were analyzed after 6 months (T1). The authors detected highly significant protein differences: 30 protein spots between T0 and T1, 39 between healthy control and patient group at T0, and 30 between controls and patients at T1. An interesting finding of this 2D PAGE analysis was a tendency for the CSF protein pattern of SMA patients to revert toward that of control donors after treatment. Yet, the protein profile of the T1 CSF was closer to T0 SMA CSF than to control CSF. Differential expression of proteins was observed after treatment. As compared to Kessler et al. [29], the pattern of proteins was different after treatment.

The researchers then used MALDI-TOF MS to identify significantly different protein spots on MS preparative gels and identified, among 30 excised protein spots, 17 of them, representing 6 unique proteins unambiguously characterized. They ultimately examined the GO terms for six differentially expressed proteins in UniProtKB in order to determine their biological meaning.

### 2.15. Apolipoproteins

Apolipoprotein A1 (APOA1) and apolipoprotein E (APOE) were significantly upregulated after treatment in the direction of control level, and transthyretin (TTR) level also varied consistently in the same direction. Specifically, CSF samples from SMA and controls showed significant differences in APOE protein spots. However, before and after nusinersen administration, their spot abundance values were very similar. APOE expression was detected as increased, upon nusinersen treatment in monodimensional PAGE and Western blots.

In contrast, both 2D gel image analysis and 1D PAGE with Western blot detected an increase in APOA1 proteoforms associated with therapy.

The multifunctional proteins APOA1 and APOE constitute the primary lipoproteins in the CSF and play a crucial role in cholesterol transport [66,67]. They also regulate lipid homeostasis in plasma and tissues [66,67]. In the CNS, APOE is produced predominantly by astrocytes and to some extent microglia. In addition, neurons express APOE in response to excitotoxic injury [68,69,70,71]. Given that these proteins play key roles in processes that appear to be dysfunctional in SMA, including synapse formation, axonal elongation, neuronal survival and plasticity, inflammation, and modulation of oxidative stress, the induction of these proteins by nusinersen could represent a parallel and synergistic positive effect associated with the increase in SMN protein.

APOE is a multifunctional protein that plays important roles in lipid metabolism, including lipid transport in CSF and plasma, and regulation of its expression and gene polymorphisms are associated with neurological and neurodegenerative diseases, including AD, PD, and stroke [72]. APOE e4 is the most strongly established genetic risk variant for late-onset Alzheimer’s disease (LOAD) and APOE plays a critical role in the genetic etiology of AD also because of gene dysregulation. In AD, modifying the e4 isoform or reducing APOE levels is a potential approach of precision medicine. However, it is unclear whether this process can be applied to other neurodegenerative diseases [73]. Moreover, the CSF levels of APOA1 and APOE are reduced in patients with AD, suggesting that this is related to neurodegeneration [70]. However, another study reported that CSF APOE levels correlate with increased beta-amyloid and Tau biomarkers in AD [71]. Thus, a conclusive correlation of APOE level and neurodegeneration cannot be drawn.

### 2.16. TTR

This protein was found increased in SMA patients after treatment. TTR, known also as pre-albumin, is a homo-tetrameric protein present in plasma and CSF and it is responsible of transporting thyroxine and retinol binding proteins to different organs. The interpretation of the elevated level of TTR in SMA patients after therapy is uncertain because it is has also been reported as upregulated in patients with neurodegenerative disorders [74]. It may have a positive effect by promoting myelination [75,76]. TTR mutations result in transthyretin amyloidosis (ATTR) and that these mutations may also affect neurons and neurite outgrowth [77]. Nonetheless, TTR performs also neuroprotective functions in addition to contribute to vascular episodes, critical for AD. Indeed, the amyloid binding protein TTR interacts with amyloid Aβ peptides and has been proposed to have a neuroprotective role in AD. As TTR plays an important role in neurodegenerative diseases, targeting this protein is a promising avenue for developing new treatments [78].

### 2.17. Overall, Lipoproteins and TTR May Be Useful Biomarkers for Monitoring Patient Response and Disease Progression

#### Carbonyl Group Patterns

In order to determine whether nusinersen therapy affects also oxidative stress, Bianchi et al., [30] monitored protein carbonyl group as indirect indicators of protein oxidation in the CSF of two controls and two patients at baseline and at 6 months after treatment. The signal intensity of oxidized proteins was reduced after nusinersen treatment, likely because the drug might have a non-MN autonomous effect on the inflammatory response and thus reducing the oxidative stress.

### 2.18. Combination of Untargeted and Targeted Proteomics of the CSF of SMA Subjects before and after Treatment

In a study published in 2022, Shorling et al. [31] applied MS-based proteomic analyses to identify potential biomarkers in CSF samples from SMA patients treated with nusinersen. Using Western blot and enzyme-linked immunosorbent assay, they confirmed the profile of candidate proteins identified in CSF samples from SMA patients and in parallel measured the levels of heavy and light neurofilament chains. The authors [31] took a two-step approach. Using untargeted proteomic analysis, they first identified proteins as potential CSF biomarkers in longitudinal samples from three patients with SMA type 1 and collected on day 1 (initiation of therapy), day 14, and day 180.

Cathepsin D (CTSD), a lysosomal protease involved in protein degradation in skeletal and heart muscles, was pointed out as potential biomarker for SMA, as upregulated compared to non-SMA subjects. CTSD is an aspartyl lysosomal protease involved in the lysosomal degradation of proteins [79]. Remarkably, it is known to degrade proteins associated with neurodegenerative diseases, such as α-synuclein, amyloid precursor protein, and tau. Thus, because of its degradative function, CTSD is a promising target for therapeutic strategies to counteract the accumulation of protein aggregates in neurodegenerative diseases [80].

Subsequently, in Shorling’s study, after identifying CTSD as a potential biomarker, a validation analysis was performed in a larger cohort of 31 treated SMA pediatric patients (12 type 1, 9 type 2, 6 type 3, and 4 presymptomatically treated SMA subjects). Samples were from days 1, 60, and 300 post treatment. Active CTSD levels were significantly higher in SMA subjects, ≥2 months old, compared to non-SMA subjects. Of note, the researchers also found downregulation of CTSD in muscle biopsies from SMA patients.

After nusinersen administration, a greater reduction in CTSD levels of CSF samples was detected in “treatment responders” than in “nonresponders”. Overall, this study suggests that CTSD levels may be useful as a biomarker for SMA, particularly in older patients, if analyzed together with neurofilament light chain in adolescents or alone in adults.

## 3. Neurometabolic Patterns in CSF before and after Treatment of SMA

### 3.1. ^1^H-NMR Metabolomes of Three Samples (Urine, Serum, and CSF)

To identify the metabolites participating in the SMA cascade of biological events, a number of innovative metabolomic methods can be used, revealing details about the disorders’ molecular features and metabolic mechanisms [81].

Nuclear magnetic resonance (NMR) spectroscopy and MS are the main tools used. NMR is currently used to analyze complex matrixes, such as biological samples, in addition to target molecules [82].

Deutsch et al. [32] analyzed the effect of nusinersen on the ^1^H-NMR targeted metabolomes of three types of specimens (urine, serum, and CSF) in a cohort of male and female pediatric patients with all three types of SMA (*n* = 25) compared with matched healthy individuals (Figure 6).

Urine was chosen for its accessibility in monitoring the general health status of the patient. This single-center study examined the urine of 25 people with SMA before and after 2 months of nusinersen treatment and compared them with those from a healthy cohort of individuals (*n* = 125). Analysis was performed on a 600 MHz ^1^H-NMR spectrometer (Agilent Technologies, Santa Clara, CA, USA).

Although after nusinersen therapy the movement, posture, and strength of patients showed significant improvements, the metabolomes of urine, serum, and CSF were not significantly different before and after treatment.

Given the congruence of the results among the three biofluids, nusinersen therapy did not result in a significant change in the metabolite profile, which is quite remarkable. Likely, these results can be attributed to the reduced sensitivity of NMR compared to MS, whose sensitivity is in the nanomolar range. Therefore, it could be worth investigating SMA metabolomes using MS. However, the window chosen to ascertain metabolic changes after treatment might have been not suitable as well.

With respect to the healthy cohort, SMA ^1^H-NMR metabolomes showed significantly lower concentrations of urine metabolites (58% of those found in the healthy cohort). Moreover, SMA intra-samples variability was much larger than in matched healthy controls. Overall, SMA and healthy individuals exhibited significantly different metabolic profiles, a finding that must be investigated in future studies.

The urinary creatinine level was confirmed to be the best indicator for discriminating between SMA patients and healthy subjects by machine learning. The creatinine values were lower in SMA patients than in controls, but they did not significantly increase after nusinersen therapy. Creatinine variation in SMA may be linked to diabetes and inactivity [83] and indeed in Deutsch et al. the pattern of creatinine in SMA patients paralleled that of individuals with motor inactivity or muscle wasting. The determination of urine creatinine levels may allow simple, non-invasive monitoring of disease and muscle activity in pediatric SMA patients. Notably, two recent studies reported increased serum levels of creatinine in SMA type 2 and 3 patients after therapy [84,85].

Overall, Deutsch et al. [32] provide the first evidence that it is possible to analyze metabolomes of biological fluids in SMA patients using NMR and to distinguish them from healthy controls.

### 3.2. NMR Spectroscopy: A Second Study

Similarly, Errico et al. [33] investigated the metabolic effects of nusinersen on pediatric SMA patients (all three types, *n* = 27) using NMR spectroscopy (NMR, 600 MHz ^1^H-NMR spectrometer, Bruker Co, Rheinstetten, Germany). No healthy control group was included in the study. The biochemical effects of nusinersen were characterized by modulation of amino acid metabolism, with downstream metabolic consequences dependent on the severity of the disease. SMA severity was linked to different basal CSF metabolomes, which may be related to the patients’ diverse clinical manifestations but no biochemical metabolic differences attributable to nusinersen were detected in pooled analysis.

However, Errico et al. [33] were able to identify specific metabolic patterns associated with treatment when patients were stratified according to disease severity. The metabolic patterns presented by the three types of SMA differed in the pathways involved in energy homeostasis (ketone body turnover, Warburg effect, and amino acid metabolism). Interestingly, CSF creatinine levels inversely correlated with the severity of SMA, indicating its possible utility as a biomarker for SMA, as suggested by Deutsch et al. [32].

The glucose metabolism was activated by nusinersen in patients with severe SMA type 1. Additionally, after treatment, the CSF levels of glucose, lactic acid, and pyruvic acid increased in SMA type 1 individuals. In addition to an impact on energy homeostasis, nusinersen promoted the breakdown of valine, leucine, and isoleucine, which can result in the production of acetyl-CoA and support mitochondrial activity. Alterations in glucose and fatty acid metabolism have been suggested to cause dysregulated energy homeostasis in SMA patients.

The fasting tolerance of SMA patients is poor, resulting in hypoglycemia and ketoacidosis. Due to a boost in glucose metabolism and ATP production, nusinersen may alleviate metabolic abnormalities in severe SMA type 1. However, it is necessary to investigate the possible confounding effect of enteral feeding or gastrostomy on metabolic changes in SMA type 1.

The impact of nusinersen on SMA type 2 patients was different, as it did not alter blood glucose or pyruvic acid levels, but increased acetone, acetoacetate, and 3-hydroxybutyrate. Importantly, during fasting, ketone bodies absorb hydroxyl radicals and reduce reactive oxygen species production in the CNS. Whether fasting preceding lumbar punctures (>6 h) affected the ketone body metabolism in the CSF of SMA type 2 patients who received nusinersen is unclear. However, this is unlikely because nusinersen did not impact ketones in the CSF of patients with SMA type 3, who underwent the same fasting regimen as patients with SMA type 2.

Errico et al. [33] also found that nusinersen modulates amino acid metabolism in patients with milder SMA type 3. The metabolic effects of nusinersen in the CSF of patients with SMA type 3 were significantly different from those in patients with more severe types of SMA. The biochemical changes were primarily related to the metabolism of amino acids. Amino acids play a variety of roles in biochemical pathways, and nusinersen may be able to alter metabolic pathways differently in each type of SMA.

Nusinersen was also able to modify the CSF metabolomic profile of the more severe cohort making it similar to that of the milder one. The nusinersen-induced neurometabolic profiles in SMA type 1 patients became close to the profiles of SMA type 2 patients during the maintenance phase according to a hierarchical clustering analysis. Further, the post-treatment CSF metabolomes of SMA type 2 patients showed a stronger correlation with the before-treatment profiles of SMA type 3 patients than those of untreated SMA type 2 patients. The results of this study provide a first indication of nusinersen’s beneficial impact on CSF in patients with SMA from a metabolic standpoint.

Moreover, peripheral organ metabolism as well appears to be modulated upon intrathecal administration of nusinersen in line with CNS-specific upregulation of SMN. Indeed, nusinersen stimulates biochemical pathways in individuals with SMA type 1, including the glucose–alanine cycle, which has direct correlations with liver and muscle function. The metabolism of ketones is also modulated in individuals with SMA type 2, suggesting systemic therapeutic effects on the liver, skeletal muscles, and adipose tissue. Another benefit of nusinersen is that it increases the recycling of ammonia in SMA type 3 patients. Metabolic imbalance and peripheral organ dysfunction have been reported in mice models of SMA and in individuals with the disease [86]. Overall, nusinersen therapy activates a metabolic regulatory loop that involves functional interactions between peripheral organs and the CNS. The findings will need to be confirmed by metabolomic analyses of other samples such as blood from a greater number of SMA patients.

All these events may be consequences of increased amount of SMN in the CNS after treatment. Indeed, SMN induction influences the synthesis of various amino acids, including tyrosine, phenylalanine, tryptophan, glutamine, serine, and histidine and CNS neurotransmission may be affected by these changes. These amino acids are required to produce catecholamines (dopamine, noradrenaline), serotonin, histamine, and excitatory neurotransmitters (e.g., glutamate, aspartate, glycine, and D-serine). Thus, nusinersen stimulates neurotransmitter synthesis and directly affects the neuromuscular system by improving MN connections with skeletal muscle and by interacting with liver and adipose systems. Due to modifications in CNS neurotransmission, the hypothalamus-pituitary axis, which regulates systemic metabolism, can also affect peripheral organ metabolism.

## 4. Conclusions and Future Perspectives

Generally, nusinersen regulates molecular pathways that are SMA specific. In SMA patients, this CNS-directed therapy modulates amino acid metabolism and stimulates bioenergetic pathways that facilitate interactions of the CNS with muscles and other peripheral tissues. As NMR-based CSF analysis has indicated that amino acid metabolism is significantly affected in SMA, particularly in the severe form, we suggest that supplementing the diet with specific amino acids could potentiate nusinersen’s therapeutic effects. If left untreated, SMA is an irreversible neurodegenerative disease that may hit multiple cellular pathways and biological processes not only in the CNS but also in the entire organism. Novel SMN-increasing therapies allow modulation of these pathways, leading to phenotypical improvement, along with systemic effects, which still need to be deciphered [86]. Proteomics or metabolomic data for the most recent approved therapies are not yet available. From this perspective, studying the biochemical signatures associated with SMA treatment using high-throughput detection systems for proteins and metabolites may help to shed light on the mechanisms of disease pathogenesis and the therapeutic response.

Overall, results from available proteomic studies yielded some useful insights. One study on nusinersen-treated SMA type 1 patients showed an upregulation of APOA1, APOE, and TTR after treatment, suggesting an impact of nusinersen on pathways, such as inflammation and oxidative stress [30]. Conversely, another study on nusinersen-treated SMA type 2 and 3 patients failed to identify changes in the proteome following treatment, but was able to discriminate between two different clusters of SMA patients which correlated with clinical response to treatment.

The first metabolomic study conducted on nusinersen-treated SMA patients failed to identify an effect on treatment on urinary metabolic profile [32]. On the other hand, a subsequent study that analyzed the CSF metabolome using NMR showed differences following treatment according to the SMA subtype [33].

Despite those mentioned, deep untargeted matched proteomic and metabolomic studies that allow the profiling of biofluids at an unprecedented scale, leading to many novel hypotheses for future investigation are still missing.

To identify biomarkers for SMA prognosis, proteomic and metabolomic studies are crucial, particularly those involving longitudinally followed and well-characterized cohorts of patients.

It is important to mention that although metabolomic and proteomic approaches are promising, many technical challenges remain. The lack of standardization among detection methods is a major caveat, especially when measuring small substances. The panels of metabolites and proteins present little overlap among published studies, suggesting a risk of low reproducibility. A standard operating procedure (SOP) is recommended to harmonize measures and reduce inter-study heterogeneity.

Further, molecular alterations and characteristic proteins will continue to be discovered as the technology of MS-based omics approaches progresses. Moreover, elucidation of molecular pathways at the single cell level will significantly benefit from the introduction of single-cell metabolomics and proteomics.

The use of multi-omics analysis, which integrates transcriptomic, metabolomic, and proteomic techniques with bioinformatic analyses, could strengthen the validity of these findings, providing meaningful information.

Overall, omics-driven systems biology analysis, by linking genotype, proteotype, metabolism, and phenotype, will lead to better therapeutics and to the implementation of precision medicine, allowing patients to receive the most appropriate treatment based on their disease course.

## Figures and Tables

**Figure 1 biomedicines-11-01254-f001:**
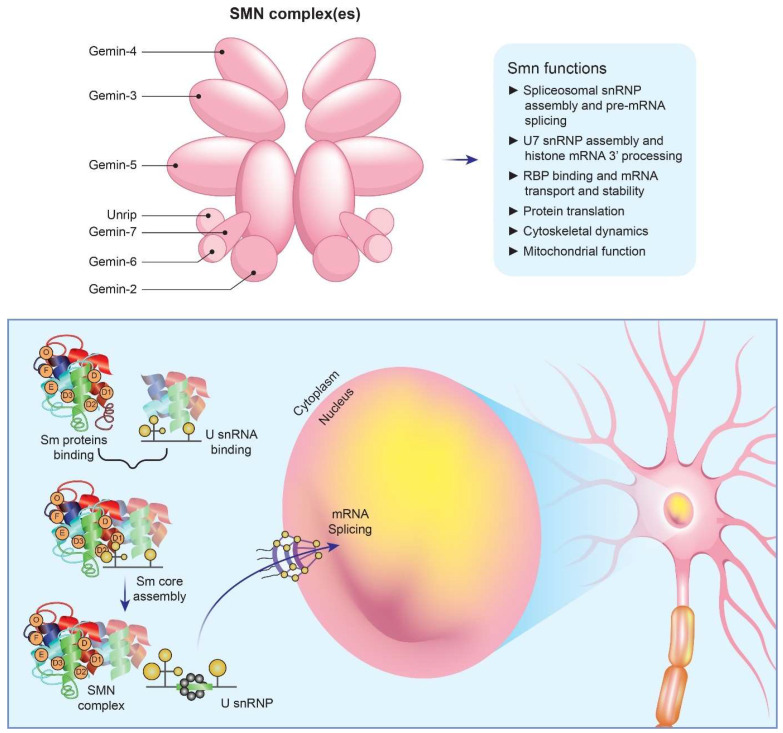
Main roles of SMN protein in cells. SMN is essential for the assembly of spliceosomal snRNPs, which are necessary for the splicing and expression of mRNAs. A cascade of events occurs as a result of SMN deficiency. Specifically, Sm core formation is impaired, SnRNP levels are decreased, stasimon is disrupted, and critical neuronal functions are disrupted, ultimately resulting in SMA.

**Figure 2 biomedicines-11-01254-f002:**
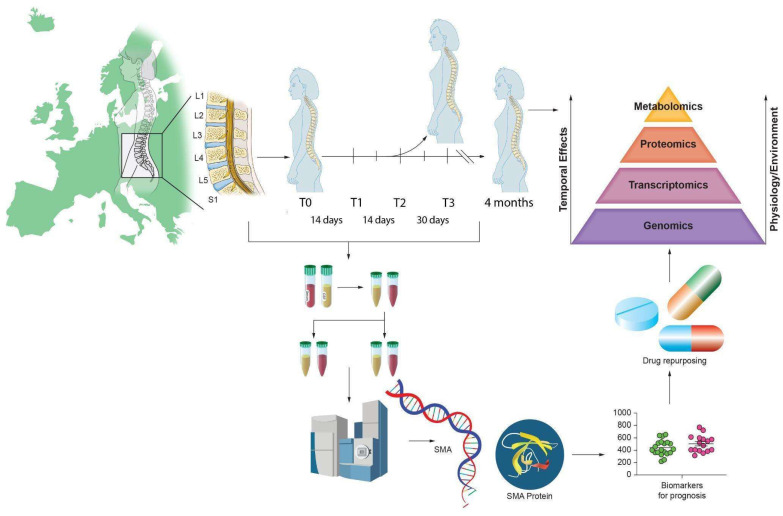
Illustrative abstract of the omics analysis paradigm to identify biomarkers in SMA.

**Figure 3 biomedicines-11-01254-f003:**
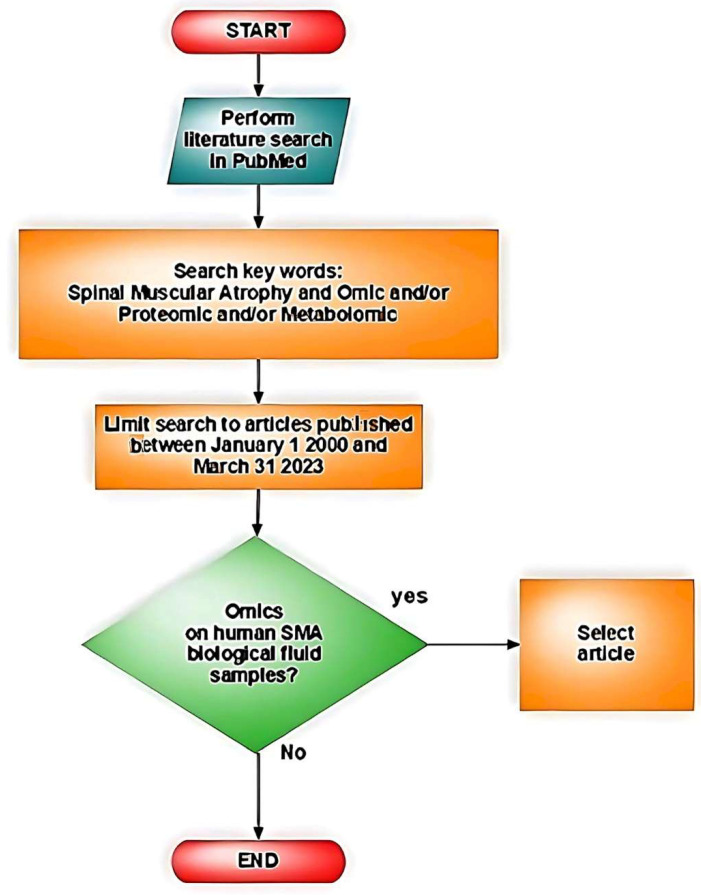
Illustrative flowchart of the literature search.

**Figure 4 biomedicines-11-01254-f004:**
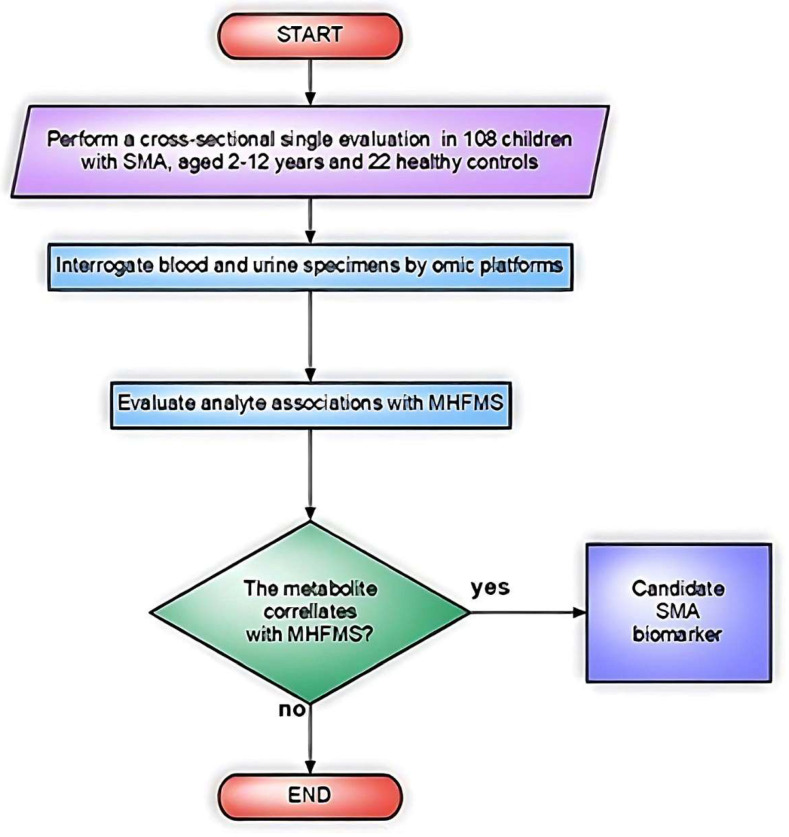
Flowchart of the study design in Finkel et al. [28].

**Figure 5 biomedicines-11-01254-f005:**
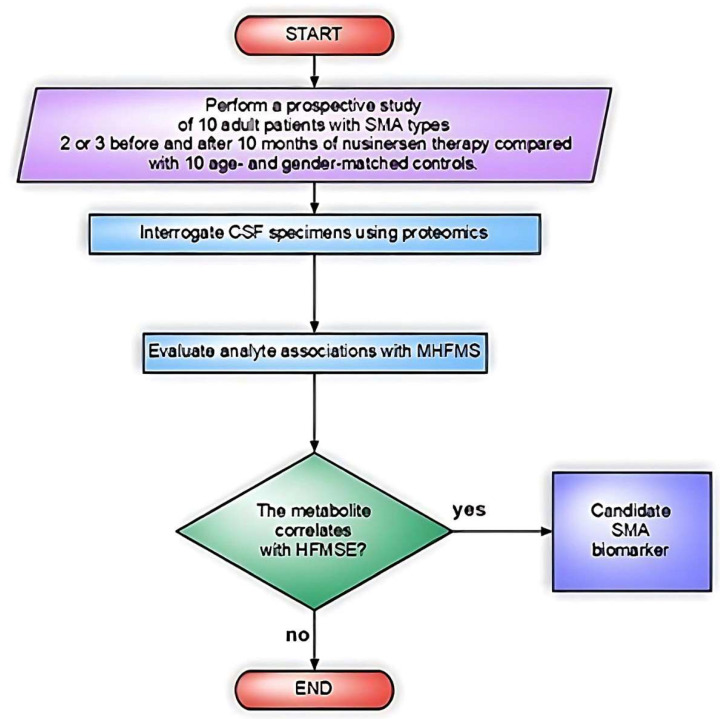
Flowchart of the study design in Kessler et al. [29].

**Figure 6 biomedicines-11-01254-f006:**
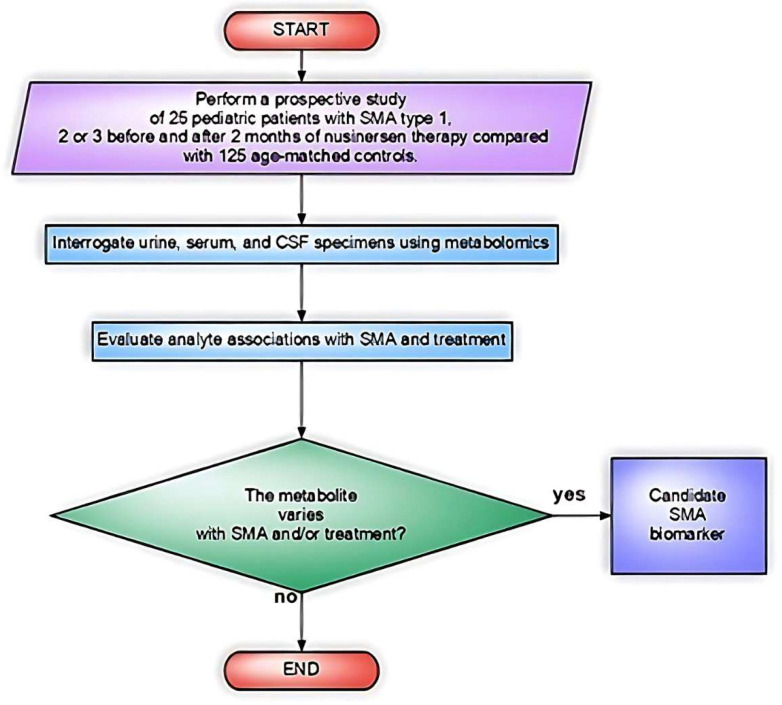
Flowchart of the study design by Deutsch et al. [32].

**Table 1 biomedicines-11-01254-t001:** Summary of the studies presented in this review.

Author, Year	Patient Cohorts	Treatment	Samples	Methods	Major Findings	Molecular Pathways
Finkel et al., 2012 [28]	108 children with SMA (type 1–3, age 2–12 years old) and 22 controls of similar age	No treatment	Plasma	Proteomics: multidimensional liquid chromatography and eight-plex iTRAQ labels with mass spectrometry. Metabolomics: QStar Elite quadrupole time-of-flight instrument.	Dipeptidyl Peptidase IV osteopontin, and tetranectin positively correlate with MHFMS; Fetuin-A and Vitronectin negatively correlate with MHFMS.	Dipeptidyl peptidase IV: glucose/insulin metabolism, immune response. Osteopontin: bone metabolism, immune response. Tetranectin: bone metabolism. Fetuin-A
Kessler et al., 2020 [29]	SMA type 3 (*n* = 9); SMA type 2 (*n* = 1) and age- and gender-matched controls (*n* = 10)	Baseline and after nusinersen treatment	CSF	Proteomics: Thermo Scientific QExactive HF-X mass spectrometer	Increased PARK7/DJ-1 in the CSF; two gene expression clusters in SMA patients based on age and two gene expression clusters in SMA patients based on treatment response. No significant single protein shift after therapy.	PARK7/DJ-1: DJ-1 acts as a chaperone and sensor under oxidative stress, protecting neurons by inhibiting α-synuclein aggregation.
Bianchi et al., 2021 [30]	SMA type 1 infants (*n* = 10) compared with age-matched controls (*n* = 10)	Baseline and after nusinersen treatment	CSF	Proteomics: SDS-PAGE followed by MALDI-TOF MS analysis	Increased levels of APOA1, APOE, and TTR after treatment.	APOA1: the primary apolipoprotein found in high-density lipoproteins (HDLs). APOE: protein linked to neurological and neurodegenerative diseases, with lipid transport, gene polymorphisms, and expression regulation being key factors. TTR: transport protein carrying thyroid hormone and vitamin A-bound retinol, with potential roles in neuroprotection and neurite outgrowth, mutations cause amyloidosis.
Schorling et al., 2022 [31]	SMA types 1, 2, and 3 (*n* = 3 mass spectrometry, validation *n* = 31)	Baseline and after nusinersen treatment	CSF	Mass spectroscopy	Cathepsin D decreased in SMA patients aged ≥2 months at the start of treatment but was only significant in patients who demonstrated a positive motor response.	Cathepsin D is a type of aspartyl protease prominently found in the central nervous system and skeletal and cardiac muscle. Its primary function is to catalyze the degradation of proteins within lysosomes.
Deutsch et al., 2021 [32]	SMA pediatric (type 1–3; *n* = 25) and matching controls (*n* = 25 serum, CSF) or *n* = 125 (urine])	Baseline and after nusinersen treatment	Urine, serum, and CSF	Metabolomics: 600 MHz ^1^H-NMR spectrometer	No metabolome shift after therapy. Creatinine urinary level reduced in SMA patients.	
Errico et al., 2022 [33]	SMA pediatric patients (all three types, *n* = 27); no control group	Baseline and after nusinersen treatment	CSF	Metabolomics: 600 MHz ^1^H-NMR spectrometer	Effects after nusinersen treatment: increased metabolic rate of glucose (SMA type 1), increased ketone levels (SMA type 2), increased amino acid metabolism (SMA type 3).	

## Data Availability

Data sharing is not applicable.

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
