# Peer review of "Identification of Novel Biomarkers of Spinal Muscular Atrophy and Therapeutic Response by Proteomic and Metabolomic Profiling of Human Biological Fluid Samples"

_biomedicines, 2023, doi:10.3390/biomedicines11051254_

Round 1
Reviewer 1 Report
In the present article, authors mainly sumarised the proteomic and metabolic profiling of biofluids after Nusinersen based treatment of spinal muscular atrophy. However they should also focus on the studies related to proteomic and metabolic profiling of other treatment strategies for SMA. Also the data should be better represented in form of tables/flowcharts/diagrams for easy and better understanding of readers. The pathohysiology related to SMA can also be elaborated diagrammatically. Review methodology, time frame and inclusion and exclusion crieteria for the selection of data should also be included in the form of flowchart. The collected data can also be discussed in a betterway. It seems that they only focus on the basics of proteomic and metabolic (OMICS) studies in conclusion however they should focus on important findings of their study (various biomarkers related to metabolic and proteomic profiling of biofluids in the different treatment strategies for SMA).
Author Response
Reviewer #1
Q1.1: In the present article, authors mainly summarized the proteomic and metabolic profiling of biofluids after Nusinersen based treatment of spinal muscular atrophy. However they should also focus on the studies related to proteomic and metabolic profiling of other treatment strategies for SMA.
A1.1: We appreciate this suggestion to include studies related to proteomic and metabolic profiling of treatment strategies other than nusinersen for spinal muscular atrophy (SMA). However, as far as we know, such omics studies are not yet available for other treatment options such as gene therapy or risdiplam. Therefore, in our article, we focused on summarizing all of the published studies that have investigated metabolomic and proteomic changes in SMA patients without any treatment or both before and after nusinersen treatment. We included the paper by Schorling et al. (2022) to the references.
Q1.2: The data should be better represented in the form of tables/flowcharts/diagrams for easy and better understanding of readers.
A1.2 Thank you for your valuable suggestion regarding the presentation of our data. The revised manuscript includes more figures, flowcharts, and diagrams to improve readability. We hope that these modifications have made it easier for to understand the research summaries.
Q1.3: The pathohysiology related to SMA can also be elaborated diagrammatically.
A1.3 We included a diagrammatic representation of the pathophysiology related to SMA in the revised manuscript. It will help to clarify how the disease develops. We believe that this change has improved the work's quality and increased its accessibility to a larger audience.
Q1.4: Review methodology, time frame and inclusion and exclusion criteria for the selection of data should also be included in the form of flowchart.
A1.4 In the revised manuscript, we included a flowchart of our review methodology, which should provide a clearer understanding of our selection criteria for readers. We hope that this modification has improved the transparency and clarity of our synthesis.
Q1.5: The collected data can also be discussed in a better way. It seems that they only focus on the basics of proteomic and metabolic (OMICS) studies in conclusion however they should focus on important findings of their study (various biomarkers related to metabolic and proteomic profiling of biofluids in the different treatment strategies for SMA).
A1.5 A more detailed discussion of the identified biomarkers was added to the revised manuscript, as well as an explanatory column in the table and their potential implications as treatment strategies for SMA. We think that this modification has improved the quality and relevance of our research sumamry and hope that it better addresses the concerns of the rewiever.
Reviewer 2 Report
A schematic summary of the studies presented in this review should be separated according to subsections. And each section should also contain more detailed information about the reported research studies.
It is not necessary to repeat the methods information in the main text that has been listed in the schematic summary.
Author Response
Reviewer #2
Q2.1 A schematic summary of the studies presented in this review should be separated according to subsections. And each section should also contain more detailed information about the reported research studies. It is not necessary to repeat the methods information in the main text that has been listed in the schematic summary.
A2.1 We appreciate the thoughtful review and valuable feedback regarding our manuscript. We thank you the reviewer for suggesting the inclusion of schematic summaries of some of the studies presented in our review. We have revised the manuscript as suggested and ensured that each sub-section contains clear information about the reported research studies. We hope that these modifications have improved the quality of our work and addressed the raised concerns.
Reviewer 3 Report
In this manuscript the Authors reviewed early and more recent literature studies regarding novel non-targeted omic strategies as useful tools in the clinical setting for patients with SMA. Specifically, the manuscript discuss data from proteomics and metabolomics analysis in serum and CSF SMA patients pre that can provide insights into molecular events underlying disease progression and treatment response.
The issue addressed in this manuscript is of interest in the field of SMA. The Authors provided a critical discussion on this theme.
However, from the reviewer’s point of view, minor revisions, below reported, are needed to improve the article and make it acceptable for publication.
Minor revisions:
-In the Introduction the Authors should discuss literature on the biomarkers, such myomiRs and cytokines, that have already identified in serum and CSF SMA patients pre- and post-treatment (i.e. Bonanno et al 2022; Bonanno et al., 2022);
-In the table 1 the Authors should include, in the "patient cohort column" or in a new column, data regarding the patients treated and non-treated.
- In general the manuscript is difficult to understand because some of the sections are currently very dense and not that easy to follow. Please therefore ensure that there is a clear logical flow throughout and that the respective molecular pathways are emphasised. This will not only make it easier for the reader to follow your line of argumentation, but also help you to convince the reader of the relevance and potential value of the omic biomarkers.
-Moreover, please also keep in mind that the Review are aimed at a wide audience, including non-specialists so it is important that all important aspects of the topic are well introduced and explained. To that end, I felt that the reader would benefit from further details regarding SMN function and their interaction with cellular pathways, including any relevant signalling pathways, and how these are altered in SMA.
Author Response
Reviewer #3
Q3.1 In the Introduction the Authors should discuss literature on the biomarkers, such myomiRs and cytokines, that have already identified in serum and CSF SMA patients pre- and post-treatment (i.e. Bonanno et al 2022; Bonanno et al., 2022);
A3.1 We thank the reviewer for taking the time to review our manuscript and for providing valuable feedbacks. As recommended, we included a discussion of the relevant literature on myomiRs and cytokines in SMA patients in the revised manuscript. Specifically, we have included the studies by Bonanno et al. (2020, 2022) that identified myomiRs and cytokines in serum and CSF from SMA patients pre- and post-treatment. We hope that these changes address the concerns and that the revised manuscript is suitable for publication.
Q3.2 In the table 1 the Authors should include, in the "patient cohort column" or in a new column, data regarding the patients treated and non-treated.
A3.3 We agree that adding a new column with treatment status information would be valuable. We revised the table to include this information and make it clearer for the reader. We appreciate the attention of this reviewe to the details and are grateful for the feedback.
Q3.3 In general the manuscript is difficult to understand because some of the sections are currently very dense and not that easy to follow. Please therefore ensure that there is a clear logical flow throughout and that the respective molecular pathways are emphasized. This will not only make it easier for the reader to follow your line of argumentation, but also help you to convince the reader of the relevance and potential value of the omic biomarkers.
A3.3 We took the advice of the reviewer to ensure that there is a clear and logical flow throughout the paper, and to emphasize the respective molecular pathways that we are discussing. We also provided more context and explanations for the terms and concepts that we describe in the paper. We agree that it is important to make our research accessible and understandable to a wide range of readers, and we attempted to present our findings in a way that is both informative and engaging. Regarding the molecular pathways, we added information in the text and table.
Q3.4 Moreover, please also keep in mind that the Review are aimed at a wide audience, including non-specialists so it is important that all important aspects of the topic are well introduced and explained. To that end, I felt that the reader would benefit from further details regarding SMN function and their interaction with cellular pathways, including any relevant signalling pathways, and how these are altered in SMA.
A3.4 We acknowledge the importance of providing a clear and comprehensive account of the molecular pathways involved in SMA and the role of SMN in these pathways to make the manuscript accessible to a wider audience, including non-specialists. Accordingly, we have made several updates to the manuscript, including a more extensive introduction to the topic, an explanation of SMN's function, its interaction with cellular pathways, and relevant signaling pathways that are altered in SMA. We have also added an explanatory figure to aid in understanding. Additionally, we have ensured that the information is presented in a clear and concise manner that is accessible to readers with different backgrounds.
Round 2
Reviewer 1 Report
Nil